# Subclinical Left Ventricular Dysfunction over Seven-Year Follow-Up in Type 2 Diabetes Patients without Cardiovascular Diseases

**DOI:** 10.3390/biomedicines12092031

**Published:** 2024-09-05

**Authors:** Dariga Uaydinichna Akasheva, Tatyana Gennadyevna Utina, Olga Nikolaevna Dzhioeva, Oxana Mikhailovna Drapkina

**Affiliations:** National Medical Research Center for Therapy and Preventive Medicine of the Ministry of Healthcare of the Russian Federation, 101000 Moscow, Russia; tutina@gnicpm.ru (T.G.U.); odzhioeva@gnicpm.ru (O.N.D.); odrapkina@gnicpm.ru (O.M.D.)

**Keywords:** type 2 diabetes, subclinical left ventricular dysfunction, diastolic disfunction, global longitudinal strain

## Abstract

Subclinical left ventricular dysfunction (LVD) is common in asymptomatic patients with type 2 diabetes (T2D). This study aimed to define long-term structural and functional disorders of the left ventricle (LV) myocardium over a 7-year follow-up in patients with T2D without cardiovascular diseases (CVD). Of the 120 patients with and without T2D of both sexes aged from 45 to 75 years (57.11 ± 7.9 years), included in the study in 2012–2013, 57 responded to the follow-up study. They were divided into two groups: one with T2D (*n* = 29), the other without it, the control (*n* = 28). All patients underwent transthoracic two-dimensional echocardiography with an assessment of standard indicators of systolic and diastolic cardiac function, global longitudinal strain (GLS), laboratory diagnostics of carbohydrate metabolism disorders markers, NT-proBNP, and CRP. The median follow-up duration was 7.2 [7.0–7.8] years. During the follow-up, a statistically significant increase in the incidence of diastolic dysfunction (DD) from 53% to 61% (*p* = 0.004) was found in the T2D group; no significant dynamics were noted in the control group (*p* = 0.48). The proportion of patients with reduced GLS (<−18%) increased in the T2D group (*p* = 0.036). A significant difference in the frequency of decreased GLS depending on presence of T2D was demonstrated. In conclusion, T2D is an independent risk factor for the worsening of subclinical left ventricular dysfunction in asymptomatic patients with T2D without CVD over 7-year follow-up.

## 1. Introduction

Today, one in ten people globally lives with diabetes, amounting to more than half a billion adults (537 million). By 2045, this number is projected to rise by 46%, reaching 783 million people (one in eight) [1]. In parallel with the increasing prevalence of diabetes, there is a corresponding rise in its major complications, including heart failure (HF). The key link between diabetes and HF is believed to be diabetic cardiomyopathy (DCM). DCM is commonly defined as structural and functional myocardial disorders that are driven by metabolic abnormalities associated with diabetes, rather than by known cardiovascular disease (CVD) risk factors such as hypertension, dyslipidemia, coronary artery disease, or valvular disease [2,3]. DCM still remains a clinical concept without its own code in the international classification of diseases. It can occur in any form of diabetes; however, most research on diagnosis has focused on patients with type 2 diabetes (T2D) [4]. Specific diagnostic criteria for DCM, whether clinical, instrumental, or laboratory-based, have not yet been established. Consequently, the reported prevalence of DCM in patients with T2D varies widely, ranging from 5% to 58% [2,4,5,6]. The concept of DCM is broad and encompasses a wide range of phenotypes, including those with reduced and preserved ejection fraction (EF) of the left ventricle (LV). In its early stages, DCM is often asymptomatic and is considered a form of stage B HF (SBHF), associated with an increased risk of progressing to symptomatic stages of overt HF [4,6,7,8,9]. The term “subclinical left ventricular dysfunction” (LVD) has been introduced to describe this subclinical condition and facilitate early screening for initial DCM [10]. Subclinical LVD thus represents a transitional phenotype that precedes the onset of symptomatic HF.

Echocardiography is the most commonly used method to examine cardiac function in patients with diabetes. In 23–70% of asymptomatic patients with diabetes, LV diastolic dysfunction (DD) is detected using conventional echocardiography and tissue Doppler imaging [11,12]. Even with a normal EF, patients with T2D often show evidence of subclinical systolic dysfunction, as indicated by a lower global longitudinal strain (GLS) [13,14]. Moreover, a reduced GLS in absolute value (<–18%) is likely the earliest ultrasonic sign of preclinical DCM [15,16]. The presence of subclinical LVD in asymptomatic patients with T2D is independently associated with adverse outcomes over long-term follow-up [11,17,18,19]. A systematic review and meta-analysis of 11 distinct studies demonstrated that both subclinical systolic and diastolic LVD are associated with a substantial risk of developing HF, highlighting the urgent need for effective interventions at these stages [20].

DCM has been studied for over 50 years, beginning with Rubler’s seminal paper [21], with research activity intensifying over the past two decades, as confirmed by a comprehensive systematic bibliometric analysis [22]. Despite the significant accumulation of knowledge, DCM remains a concept with an ill-defined phenotype and an unclear picture of its progression. The time required for the transition from asymptomatic to manifested HF, and the factors influencing this transition, are not fully known. To date, no study has comprehensively tracked the full history of DCM, from subclinical LVD to overt HF. The ongoing debate about whether there are two distinct DCM phenotypes, HF with preserved EF (HFpEF) or HF with reduced EF (HFrEF), versus a single evolving phenotype remains unresolved. This debate is not merely academic; it fundamentally determines the tactics of preventive measures.

Thus, DCM is commonly asymptomatic and represents subclinical LVD. It is an underrecognized entity that may progress to functional decline and overt HF. Early recognition of the high-risk group for HF in patients with T2D is vital, given the evidence of poor outcomes. More observational longitudinal clinical studies and sustained additional efforts should be made to target this stage in order to decrease the overall burden of HF in diabetes. In our study, we aimed to evaluate the long-term dynamics of subclinical LVD in patients with T2D free of CVD. 

## 2. Materials and Methods

### 2.1. Study Population

A total of 120 patients with and without T2D of both sexes aged from 45 to 75 years (57.11 ± 7.9 years) were recruited in the study in 2012–2013 after screening. Screening details, as well as inclusion and non-inclusion criteria, were described thoroughly in our previous publication [13]. In brief, the study included people who had no symptoms and/or history of severe somatic diseases and CVD, including hypertension of 2 and 3 degrees, without a history of acute cerebrovascular accident, coronary heart disease, peripheral artery disease, heart failure, heart defects, or clinically significant disorders of the rhythm and conductivity of the heart. Patients with poor echocardiograph image quality were also excluded from the analysis.

Recruitment of patients for the second phase of the study took place in 2020–2021. This time coincided with the peak of the coronavirus pandemic. In this regard, only 57 patients accepted the invitation to participate in the follow-up study. They were divided into two groups: one with T2D, and the other without it, the control.

### 2.2. Study Protocol

At the first stage of the study, all patients underwent a complete physical examination with measurement of waist circumference, height, weight, and calculation of body mass index (BMI). They were interviewed to establish clinical characteristics and data on used medications including antihypertensive and anti-diabetic. Blood pressure (BP) was taken as the average of three measurements, and an electrocardiogram (ECG) was recorded. Fasting blood samples were obtained to assess clinical and biochemical tests, lipid profile, electrolytes, and renal function to determine the levels of high-sensitive C-reactive protein (hsCRP) and NT-proBNP, serum glucose, glycosylated haemoglobin (HbA1c), and homeostasis model assessment of insulin resistance (HOMA-IR). Treadmill testing was performed to assess silent ischemia and tolerance to physical activity using the BRUCE protocol and treadmill metabolic equivalents (METs). Risk factors for cardiovascular complications, such as gender, age, dyslipidemia, arterial hypertension, excess body weight, and smoking, were also assessed. All patients who met the inclusion and exclusion criteria underwent transthoracic echocardiography followed by analysis of myocardial motion in various planes using speckle-tracking echocardiography (STE). In the second stage of the study, patients underwent exactly the same measures. The primary endpoint in this study was the dynamics of functional parameters of the heart according to conventional and speckle-tracking echocardiography over the seven-year follow-up. The study protocol was approved by the local institutional review board. The written informed consent was obtained from all participants.

### 2.3. Standard Echocardiography

Both initial and repeat echocardiographic studies were performed using the same commercially available echocardiography system (Philips IE-33, The Netherlands) and, importantly, by the same experienced sonographer. A 3.5 MHz probe was used to obtain images that were digitally stored in cine-loop format. Interventricular septal dimension at end-diastole (IVSd), relative wall thickness (RWT), left ventricular mass index (LVMI), and indexed left atrial volume were measured. LV systolic function was assessed by the measurement of a 2-dimensional ejection fraction from apical four and a two-chamber views using the modified Simpson’s biplane method. To diagnose diastolic dysfunction, we used the transmitral flow and tissue Dopplerography. The E/A ratio of peaks velocities in early diastole (E-wave) and late diastole (A-wave) was calculated. Pulsed wave tissue Doppler imaging was used to measure the early diastolic velocity (e’) with the sample volume placed at the septum annulus of mitral valve. In addition, the E/e’ ratio was calculated as an estimation of LV filling pressure. The working definitions were based on the criteria of the American and European Recommendations for the Evaluation of Left Ventricular Diastolic Function by Echocardiography [23].

### 2.4. Speckle-Tracking Echocardiography

Global longitudinal strain (GLS) was measured in the 3 standard apical views in B-mode with increasing grayscale gradation at a frequency of 60–80 frames per second. In each position, 3 consecutive cardiac cycles were recorded at the end of expiration while holding the breath and stored for later analysis. In the apical 3-chamber view, the operator set the timing of closure of the aortic and mitral valves. Calculation of myocardial deformation was triggered by the R wave on the ECG. The time from the peak of the R wave to the closure of the aortic valves was calculated automatically. The endocardial border was traced manually. The resulting images were subsequently processed on a QLAB workstation (Advanced Ultrasound Quantification Software Release 8.1.2, Philips). The strain values of all LV segments were averaged and reported as GLS (%). The reduction in the GLS level was taken as the value specified by the manufacturer for this workstation and amounted to <−18% in absolute value [24,25].

### 2.5. Statistical Analysis

Statistical analysis of the data was carried out using the SPSS Statistics 26 software package. The one-sample Kolmogorov–Smirnov test was used to check the distribution for normality. Descriptive statistics for normally distributed continuous variables are presented as the mean (±standard deviation). Non-normally distributed data are expressed as the median (Me) and interquartile range (IQR): 25th and 75th percentiles. Categorical variables are reported as the count and percentage. Depending on the type of distribution of quantitative variables, the Student *t*-test or the Mann–Whitney U test was used to compare 2 independent groups. Qualitative variables were compared using the χ^2^ test and two-sided Fisher’s exact test. Depending on the type of distribution of quantitative variables, the *t*-test or Wilcoxon test was used to compare 2 related samples; qualitative variables of 2 related samples were assessed using the McNemar test.

The visualization of the incidence DD and decreased GLS depending on the presence of T2D was carried out by constructing Kaplan–Meier curves. The curves were compared using the Mantel–Cox log-rank test. To assess the role of T2D in relation to the development of DD and reduction of GLS, a multivariate with preliminary univariate analysis was performed using the logistic regression method. In addition to T2D, the multivariate analysis included all potential risk factors for the development of DD and systolic dysfunction. Before including risk factors in the model, a univariate analysis was conducted. According to its results, significant (*p* < 0.05) or close to significant (*p* < 0.1) differences between groups were obtained. Variables that showed *p* < 0.1 in the univariate analysis were included in the model. Then, in the multivariate analysis, they were step-by-step excluded from the equation using the Wald method backward, and what remained was entered into the graph. Variables with a *p* value < 0.05 were selected as independent predictors from the final model. The results of multivariate analysis are presented as odds ratio (OR) and 95% confidence interval (95% CI).

The significance of differences for all tested hypotheses was set at *p* < 0.05.

## 3. Results

### 3.1. Clinical Characteristics of Patients Included in the Study over Follow-Up

Clinical characteristics of 57 patients included in the final analysis are presented in Table 1. Their mean age at recruitment was 57.1 ± 7.9 years, and 55% of subjects were men. They were divided into two groups: the first group consisted of patients with type 2 diabetes mellitus (*n* = 29), and the second group included patients without it, the controls (*n* = 28). The groups were comparable by age and gender. Overall, the patients were overweight. Patients with T2D had higher body weight compared to the control group but not more than first-degree obesity. All participants had systolic blood pressure within the target values. Moreover, patients with T2D more often than patients in the control group took angiotensin-converting enzyme inhibitors and statins. The groups did not differ in the presence and number of risk factors for coronary heart disease (arterial hypertension, dyslipidemia, obesity, smoking). There were also no statistically significant differences in exercise capacity (mean 6.3 ± 1.9 METs in the control group and 6.0 ± 2.0 METs in the T2D group; *p* = 0.65). In patients with T2D at the time of inclusion in the study, the average duration of the disease was 3.3 ± 2.4 years. The patients did not have diabetic complications at the time of the study.

The mean follow-up duration was 7.2 [7.0–7.8] years. Over the follow-up, there was no statistically significant progression of hypertension, dyslipidemia, weight gain, obesity, smoking, and exercise capacity in both the T2D group and the control group. None of the patients developed cardiovascular complications. In patients with T2D, there was no increase in the incidence of microvascular complications (diabetic polyneuropathy, nephropathy and retinopathy). All patients had no clinical signs of heart failure.

When assessing the dynamics of laboratory parameters in patients with T2D, the following results were obtained: the proportion of patients with elevated HbA1c levels decreased from 67% to 47% (*p* = 0.03), but no statistically significant changes in fasting glycemia occurred. Insulin levels increased in 79% of patients (*p* = 0.005), and the HOMA index in 79% (*p* = 0.006). The level of NT-proBNP increased by 68.4 (*p* = 0.004). A decrease in hsCRP levels was also detected in 68% of patients with T2D (*p* = 0.009). But the level of both markers remained within normal limits. In the control group, no statistically significant changes in laboratory parameters were detected.

### 3.2. Echocardiography

On study entry, slightly more than half of the patients with T2D (53%) and one-third of control patients (32%) had LVD detected by echocardiography. The numbers of participants with abnormal systolic, diastolic and LV geometry parameters are shown in Table 2. This table also provides a comparison of echocardiographic parameters over a 7-year follow-up period in two groups depending on the presence T2D. Among the patients with T2D, there was a statistically significant increase in the thickness of the interventricular septum and the posterior wall of the LV (an increase was observed in 55.6% and 56%; *p* = 0.046 and *p* = 0.03, respectively), as well as the relative wall thickness (an increase was detected in 75%; *p* = 0.06). A significant increase in the incidence of concentric hypertrophy in patients with T2D was revealed (from 21% to 53%; *p* = 0.025). In the absence of T2D, changes in these parameters were statistically insignificant.

Diastolic parameters indicated a mildly impaired LV relaxation. In patients with T2D, compared with the control group, the following statistically significant results were revealed: a lower value of the E/A ratio (*p* < 0.001), a higher value of IVRT and DT (*p* < 0.001; *p* < 0.001, respectively), as well as E/e’ (*p* < 0.001). When assessing LV systolic function by EF, no disturbances were detected in any of the study participants. Regarding GLS, the T2DM group showed a statistically significant lower strain value compared to the control group (*p* = 0.03).

During the observation period, a statistically significant increase in the frequency of asymptomatic LVD in the T2D group was found from 53% to 61% (*p* = 0.004); there was no significant increase in the control group (*p* = 0.48). Figure 1 presents the results of comparison of myocardial diastolic function over time depending on the presence of T2D. In accordance with the data obtained, among 29 patients with T2D, 16 (56%) showed a statistically significant decrease in E/A (*p* = 0.03) (Figure 1A) and an increase in DT (*p* = 0.04) (Figure 1B). Overall, 18 patients (61%) had a decrease in e’ (*p* = 0.03) (Figure 1C), and 19 patients (67%) had an increase in E/e’ (*p* = 0.02) (Figure 1D). In the absence of T2D, changes in these indicators were statistically insignificant.

On the subject of LV systolic function, EF was normal and did not have significant dynamics during follow-up in both groups. At the same time, the proportion of patients with a reduced GLS absolute level of less than −18% has increased in the T2D group from 45% to 72% (*p* = 0.036), in the control group there was no significant increase (*p* = 0.62).

To estimate the survival function of the time event of interest depending on the presence and absence of T2D, Kaplan–Meier curves were constructed. The events of interest were the development of diastolic (Figure 2) and systolic dysfunction in terms of decrease in the absolute value of GLS (Figure 3) over the observation period. The median period of development of DD in patients with T2D was 6.8 ± 0.4 years (95% CI: 6.2–8.3), and in patients in the control group, it was 7.5 ± 0.5 years (95% CI: 6.7–8.8). The average time to develop GLS less than −18% in patients with T2D was 7.3 ± 0.4 years (95% CI: 6.6–8.0); in the control group, it was 7.6 ± 0.2 years (95% CI: 7.4–7.9). Log-rank testing for significance between strata demonstrated that those with T2D had significantly shorter the event-free time than those without T2D for both DD and GLS (χ^2^ = 4.86, *p* = 0.030; Figure 2 and χ^2^ = 6.81, *p* = 0.009; Figure 3, respectively).

Considering various risk factors for the development of DD and using a multivariate logistic regression method, the following independent predictors were established: the presence of T2D, increased fasting glucose levels, and an increased HOMA index (Figure 4). Independent predictors for the development of GLS less than −18% (in absolute value) according to the results of logistic regression were the HOMA index, hsCRP, and obesity (Figure 5). As a result of carrying out the analysis using the Wald method backward, those variables that are shown in the graph remained. The multivariate analysis was preceded by univariate analysis, according to the results of which variables showing *p* < 0.1 were included in the model.

## 4. Discussion

Usually, it is not easy to collect all participants for the second stage of a study after several years. In our study, this problem was exacerbated by the height of the COVID pandemic. Nevertheless, after analyzing the obtained results, we decided to publish them despite the small number of study participants. So, what can our study add to what is already known?

It is known that diabetes is associated with an increased risk of HF, and DCM refers to the abnormal structure and function of the heart associated with the metabolic disorders of diabetes in the absence of other leading cardiac factors such as hypertension, coronary heart disease, and valvular heart disease [2,3]. At the initial stage, DCM does not manifest itself clinically for quite a long time [9]. This makes its recognition difficult given the lack of specific diagnostic criteria. Depending on the used criteria, according to a large pooled epidemiological cohort study, its prevalence varies widely (11.7–67%) [5]. A recent retrospective study in real-world clinical practice indicated that DCM affects a significant portion of T2D patients. About 16% of T2D patients with asymptomatic SBHF (3% of the entire T2D population) had “pure” DCM that was associated solely with T2D. The remaining 84% (at least 15% of the entire T2D population) had “mixed” SBHF that may have resulted from the coexistence of T2D and other comorbidities, such as HTN and CAD [6].

Echocardiography is a foundation for assessment of myocardial structure and function. The main echocardiographic signs of a diabetic heart are systolic, diastolic dysfunction and abnormal LV geometry [26]. Regarding LV geometry, the early stage of DCM is characterized by concentric LV hypertrophy with normal LV diameter and volume despite the fact that DCM was initially described as a dilated phenotype with eccentric remodeling and LV systolic dysfunction [21]. The estimated prevalence of LV hypertrophy is approximately 70% in adult patients with diabetes [27]. In our study, the most common type of LV myocardial remodeling was concentric hypertrophy: 22% in the T2D group compared with controls (12%, *p* = 0.01). Moreover, during the observation period, the former experienced a significant increase in the proportion of patients with concentric hypertrophy to 53% (*p* = 0.025), while there was no significant increase in the control group.

The initial stage of DCM is characterized by a concentric phenotype with a predominance of diastolic dysfunction. According to a large systematic review with a meta-analysis by Bouthoorn S. et al., the pooled prevalence of DD in men and women with T2D in the hospital population was 48% (95% CI: 38–59%), and it was 35% (95% CI: 24–46%) in the overall population. Heterogeneity of the results associated with different diagnostic criteria across studies was high in both populations, with estimates ranging from 19% to 81% in the hospital population and from 23% to 54% in the general population [28]. As a result, Grigorescu E.D. et al., having compared studies with different approaches to determining DD, found that the 2016 European Society of Cardiology definition of DD [29] was more effective not only for selecting “undetected” cases of DD among patients with T2D, but also for predicting severe cardiovascular events (using the E/e’ ratio >14) in patients with established DD [10]. In the present study, this algorithm was used based on tissue Doppler parameters in combination with indexed definition of the left atrium and the peak velocity of tricuspid regurgitation. Initially, the DD (delayed relaxation) was present in 53% of patients with T2D and 32% without diabetes [13]. Over a 7-year follow-up, the presence of T2D statistically significantly increased this prevalence to 61% (*p* = 0.004) regardless of age, HTN, dyslipidemia, and obesity. At the same time, the diastole disturbance remained at the level of first degree, that is, delayed relaxation. There were no statistically significant changes of diastolic function in the control group.

As for the systolic function, with normal LVEF, an advanced echocardiographic technique with tracking the trajectory of myocardial acoustic markers (speckles) during the cardiac cycle was used to diagnose more subtle disorders. It seems more accurate in the modern era of precision medicine. It is also especially in demand, given that the current time is marked by the predominance of HFpEF, in which EF has no prognostic value [7]. The most robust and reproducible parameter of myocardial strain is GLS [25]. It reflects the percentage change in the length of the heart muscle during the cardiac cycle. A negative value of this parameter indicates systolic shortening of the myocardium, and a higher absolute value indicates a better LV systolic function. GLS serves as a reliable parameter for assessing systolic function of the LV myocardium in HFpEF, including patients with T2D [14,30,31]. In addition, the prognostic significance of GLS was demonstrated, the reduction of which was independently associated with adverse long-term, 10-year outcomes in asymptomatic patients with T2D [17]. In our prior study, speckle-tracking analysis showed significant differences between diabetes and control groups [13]. Over a 7-year follow-up, the proportion of patients with a reduced absolute level of GLS has increased in the T2D group from 45% to 72% (*p* = 0.036); meanwhile, there was no significant increase (*p* = 0.62) in the control group.

Using Kaplan–Meier curves, we demonstrated that the presence of T2D statistically significantly shortens the life without developing or worsening LVD, both diastolic function and GLS. In a multivariate regression analysis, independent predictors of the development of DD were not only the presence of T2D, but also indicators of carbohydrate metabolism disorders (fasting glucose, glycated hemoglobin, and HOMA index) and inflammation (hsC-reactive protein). Along with the HOMA index and hsCRP, obesity was found to be an independent predictor of the development of GLS. The full pathogenesis of DCM remains unclear to this day. It is clear that the main triggers for its development are hyperglycemia and insulin resistance. Among the many pathogenetic mechanisms underlying the functional and structural abnormalities of the diabetic heart, inflammation plays a significant role [32,33]. As a result, the data we obtained on the independent relationship of DCM with indicators of glycemia, insulin resistance, and inflammation confirm the basic postulates of the pathogenesis of diabetic cardiomyopathy.

Given the high prevalence of subclinical HF in diabetes, as well as the lack of specific diagnostic criteria for DCM, the American Diabetes Association proposes using the Universal Definition of HF criteria both echocardiographic and laboratory [34]. Their consensus document recommends testing natriuretic peptides or high-sensitivity cardiac troponin in diabetes at least once a year to diagnose subclinical HF and assess its risk of progression to clinical HF. Based on the results of population and clinical studies, they recommend considering the threshold values of biomarkers for HF as the cut-off points for DCM: BNP—50 pg/mL, NT-proBNP—125 pg/mL, high-sensitivity cardiac troponin—>99th percentile [2].

But this recommendation does not seem to be flawless: the use of BNP for the diagnosis of DCM, especially subclinical, is problematic. There is increasing evidence of normal levels of natriuretic peptide in the blood in a significant proportion of patients with HFpEF [35,36,37], which does not equate to the absence of HF. It is noteworthy that the title of the editorial for one of these papers contained a mnemonic phrase with an alternative decoding of the abbreviation: BNP—Biomarker Not Perfect [38]. In the prior work of our group, we studied the relationship of structural and functional changes in the myocardium with blood biomarkers in asymptomatic patients with T2D and also assessed the possibility of using of BNP levels as a diagnostic criterion for DCM. We solved the first task by obtaining clear associations. As for second task, the insufficient level of increase of blood NT-proBNP, within the reference values (68 [35–108] pg/mL, median with interquartile range 25–75%), also demonstrated the “imperfection” of natriuretic peptides, which means the ineffectiveness of its use in the diagnosis of subclinical LVD in T2D [39].

However, this biomarker should not be completely discounted, because it can be used to identify the degree of structural and functional cardiac disorders in patients with T2D. We suppose that low levels of NT-proBNP indicate an earlier subclinical stage of DCM, characterized by initial signs of diastolic dysfunction (delayed LV relaxation), a slight decrease in GLP, and initial remodeling of the LV myocardium by type of concentric hypertrophy. Over the 7-year follow-up, the level of NT-proBNP increased statistically significantly only in the T2D group and not in the control group. This increase in the level of NT-proBNP correlates with a deterioration in DD parameters, a decrease in GLS, and LV remodeling of the concentric hypertrophy type. Thus, the increase in the levels of NT-proBNP occurred within the reference values and this correlated with the worsening of LVD, it is important to note, within the subclinical dysfunction range. In contrast, in the study of Dal Canto et al., it was demonstrated that high levels of natriuretic peptides corresponded to more advanced HFpEF with greater LV myocardial stiffness due to more severe myocardial extracellular matrix remodeling as assessed by echocardiography and cardiac magnetic resonance imaging [40].

It is clear that there is no clarity regarding the diagnosis and prevalence of DCM. Regarding the prognostic value of DCM, the vast accumulation of data demonstrates its association with incident HF and decreased survival. The study in asymptomatic patients with T2D over a 10-year follow-up found poor prognoses associated with subclinical LVD measured by GLS. The primary endpoints for this study were all-cause mortality and hospitalization [17]. The Olmsted County population-based study demonstrated that an increase in the passive transmitral LV inflow velocity to tissue Doppler imaging velocity of the medial mitral annulus during passive filling (E/e’) ratio in diabetic patients was associated with the subsequent development of HF and increased mortality independent of hypertension, coronary disease, or other echocardiographic parameters [11]. Transition to symptomatic HF is underlain by multiple factors, including both cardiovascular and noncardiovascular determinants. So, the management strategies targeting cardiovascular and systemic comorbidities in patients identified as having ALVDD may delay symptomatic progression and improve prognosis [7,11]. In the recent large prospective cohort study, it was revealed that patients with T2D represented a more severe clinical profile and experienced more adverse outcomes compared to those without T2D [18]. The contingent of participants in this study is different from ours, since it included patients with dilated cardiomyopathy and reduced LV ejection fraction. The clinical outcomes of these patients compared to our asymptomatic patients are certainly worse. Another prospective study, like ours, included patients with T2D free of CVD. But that is where the similarities between our studies ended. In contrast to ours, it included a large number of participants and obtained significant results with a large number of outcomes, including 2625 deaths (9.5%). This enabled the authors to conclude that subclinical LVD has a high disease burden in patients without established CVD, and its prevalence is higher in patients with T2D, which was associated with an excessive mortality rate. Therefore, early screening and intervention should be considered in diabetic patients [19]. Thus, subclinical LVD in patients with T2D is the defining feature of SBHF and associated not only with a high risk of progression to overt HF, but with adverse outcome in the form of increased morbidity and mortality.

In our study, we did not obtain any so-called hard endpoints including overt heart failure. We assessed the results using surrogate points, the dynamics of myocardial function (DD and GLS). We obtained significant negative dynamics of these indicators in the group of patients with T2D compared with people without it. However, none of the study participants progressed to the clinically manifested stage of HF over a 7-year follow-up. We attribute this to two reasons. The first, perhaps, is due to the fact that the study included people with uncomplicated T2DM and normal blood levels of NT-proBNP without the clinical manifestations of CVD and/or stage II-III obesity. Moreover, it can be assumed that the “healthiest” subjects responded to the offer to participate in the repeat study. Therefore, patients who refused and were lost to the study were “less healthy” and could have poor clinical outcomes, even death. However, this information is unknown to us. Another reason for the lack of poor clinical outcomes is that all patients in the repeat study were under close medical supervision. They received recommendations on a healthy lifestyle, proper nutrition, and physical activity. They had adequate antihypertensive, antihyperglycemic, and lipid-lowering therapy. According to the results of the second point, patients with T2D showed a significant decrease in the level of glycated hemoglobin. Also, participants in the stage 2 of the study did not have a significant increase in body weight, nor a decrease in exercise tolerance. This allows us to conclude that proper control of cardiovascular risk factors, including good control of glycemia, weight, blood pressure and lipidemia, can avoid adverse outcomes over a 7-year follow-up.

Despite the large accumulation of research, the essence of DCM remains unclear. Are there two independent phenotypes (restrictive and dilated) or only one in DCM? The latter seems more believable. Only one phenotype means an evolution of the same thing, a gradual progress from diastolic to systolic dysfunction accompanied by the progress from restrictive to dilated remodeling of LV. Two independent phenotypes assume two different diseases that require different management and prevention tactics. We were unable to contribute to the resolution of the issue concerning the phenotypic essence of DCM, since we did not obtain a single hard endpoint, including the development of overt HF. However, other studies have also failed to obtain a full story of the evolution of DCM, which requires further longitudinal clinical studies. At the same time, a full understanding has been achieved that initial DCM represents a transitional phenotype prior to the onset of symptomatic HF. Without early recognition of DCM at stage of subclinical LVD, it is not possible to stop or delay this transition. In the absence of specific preventive measures for HF, we recommend timely early diagnosis of subclinical LVD in diabetic patients to identify a high-risk group for HF, strict control of cardiovascular risk factors, monitoring of comorbidities and adherence to a healthy lifestyle.

The main serious limitation of our single-center study was the small number of participants. Further prospective multi-center studies with larger patient populations will be needed to assess not only our findings, but also other unclear issues in the development of heart failure in diabetes.

## 5. Conclusions

T2D is an independent risk factor for the subclinical LVD in asymptomatic patients without CVD over a 7-year follow-up. Statistically significant worsening of LVD was represented by (1) first-degree diastolic dysfunction; (2) systolic dysfunction, with preserved LVEF in terms lower global longitudinal strain (GLS); and (3) LV myocardial remodeling in the form of concentric hypertrophy. It was demonstrated by a group of patients with T2D compared with a control group without it and remained subclinical over the entire observation period. In the absence of obvious signs of HF, early diagnosis of DCM should include not only standard but also speckle-tracking echocardiography. In the absence of specific targeted prevention of HF in diabetes at present, the tight control of glycemia, weight, blood pressure, and lipidemia may delay the transition from subclinical to overt HF and become the main therapeutic goals for preventing poor long-term outcomes associated with diabetic heart disease.

## Figures and Tables

**Figure 1 biomedicines-12-02031-f001:**
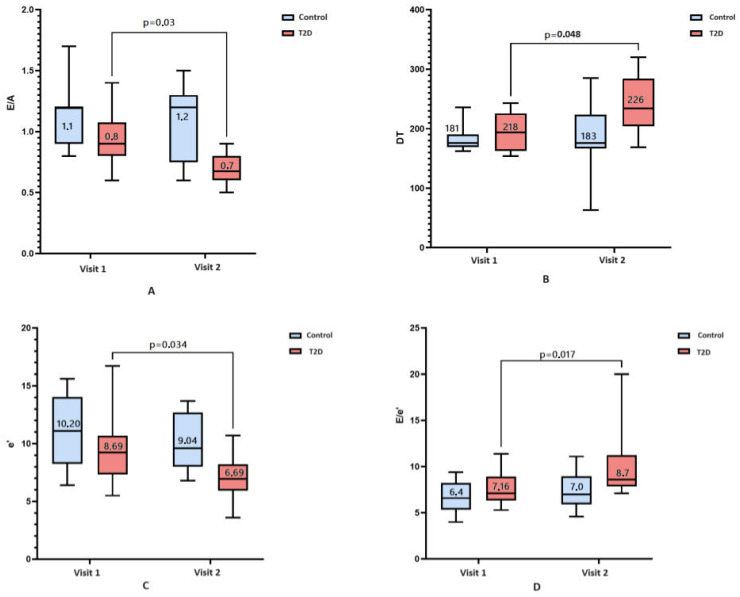
A comparison of diastolic disfunction at baseline (visit 1) and over a 7-year follow-up (visit 2) according to presence (red box) or absence of T2D (blue box). Bar charts represent: (**A**)—E/A value; (**B**)—deceleration time, ms; (**C**)—e’, cm/s; (**D**)—E/e’ value. Numeric data reported in the figure are median and interquartile range. Nonsignificant dynamics are noted in patients without T2D.

**Figure 2 biomedicines-12-02031-f002:**
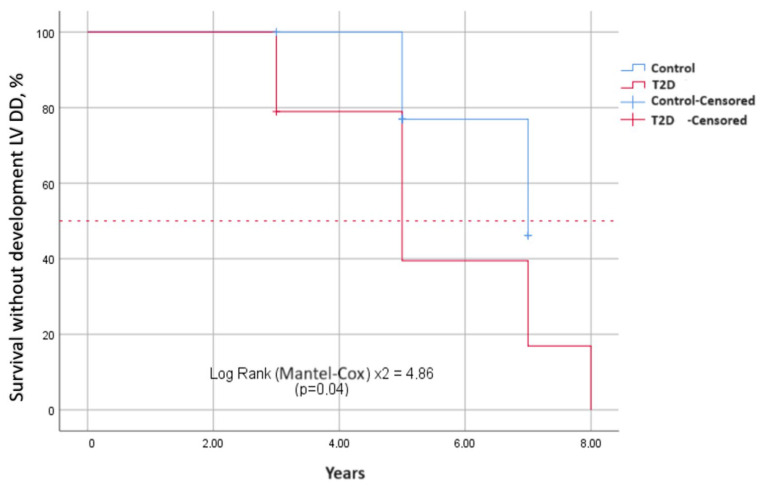
Differences in survival without diastolic disfunction between two groups depending on the presence and absence of T2D. Kaplan–Meier curves show event-free survival in years. It is longer in patients without T2D than those with T2D.

**Figure 3 biomedicines-12-02031-f003:**
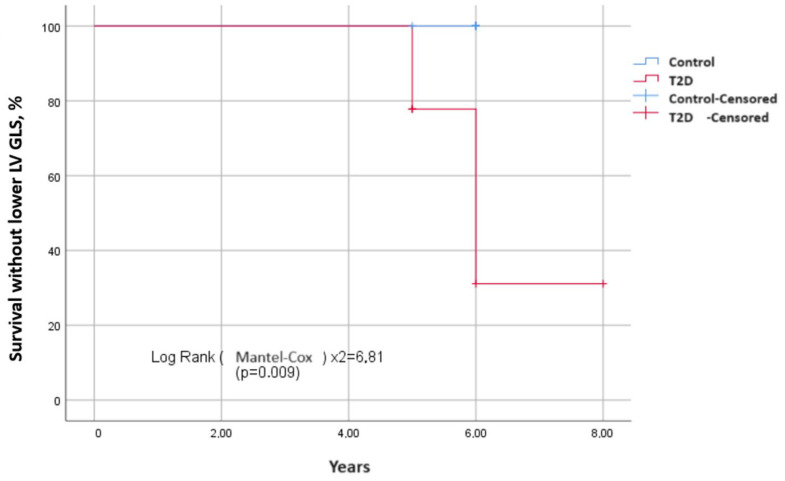
Differences in survival without the decrease of left ventricular GLS (in absolute value) between two groups depending on the presence and absence of T2D. Kaplan–Meier curves show event-free survival in years. It is longer in patients without T2D than those with T2D.

**Figure 4 biomedicines-12-02031-f004:**
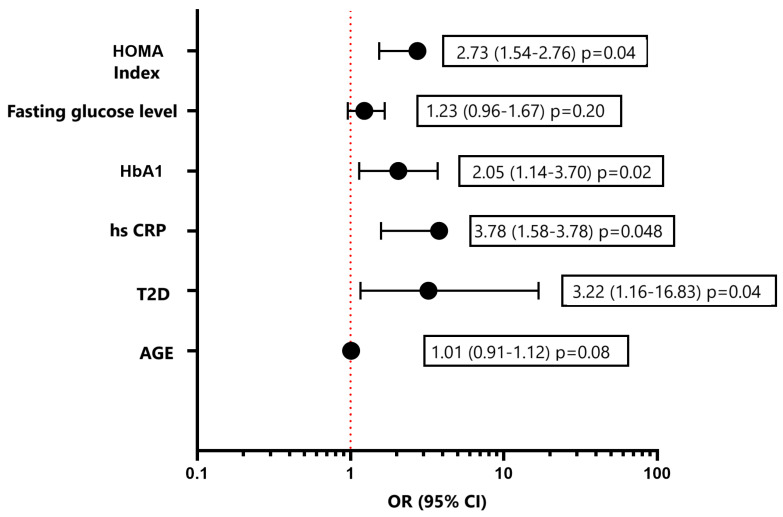
Multivariate logistic regression analysis for assessment of the risk of development diastolic disfunction LV. Forest plot of Odds Ratios for independent predictors. HbA1, glycated hemoglobin; hsCRP, high-sensitive C-reactive protein; T2D, type 2 diabetes.

**Figure 5 biomedicines-12-02031-f005:**
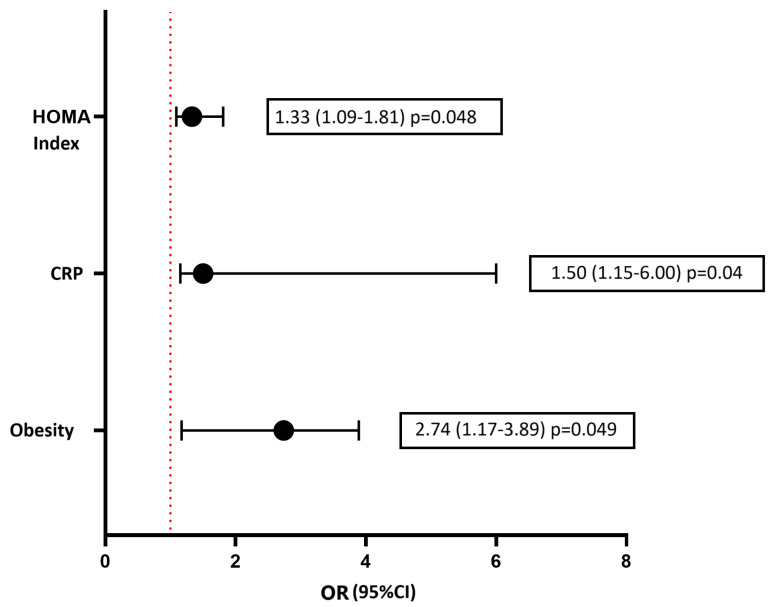
Multivariate logistic regression analysis for assessment of the risk of development the decrease of left ventricular GLS (in absolute value). Forest plot of Odds Ratios for independent predictors. hsCRP, high-sensitive C-reactive protein.

**Table 1 biomedicines-12-02031-t001:** Clinical characteristics of the study participants after 7 years of follow-up according to diabetic status.

Variable	Patients with T2D (*n* = 29)	Patients without T2D (*n* = 28)	*p* Value
Sex m, *n* (%)	13 (46.0)	12 (45.0)	0.14
Age, years, Me [IQR]	68 [58–74]	60 [55–63]	0.81
Body mass index, kg/m^2^, Me [IQR]	28 [27–33]	27 [21–28]	0.05
Obesity 1-degree, *n* (%)	13 (44.8)	3 (11.0)	0.08
Exercise capacity (METs)	6.0 ± 1.9	6.2 ± 2.0	0.59
Current smoker, *n* (%)	15 (52.0)	16 (57.0)	0.35
Creatinine, µmol/L, Me [IQR]	87 [70–91]	74 [68–94]	0.89
Cholesterol, mmol/L, Me [IQR]	4.4 [4.1–5.4]	5.6 [5.0–6.1]	0.003
Low-density lipoprotein-cholesterol, mmol/L, Me [IQR]	2.6 [1.6–3.3]	3.6 [2.9–4.1]	0.009
High-density lipoprotein-cholesterol, mmol/L, Me [IQR]	1.3 [1.2–1.4]	1.5 [1.3–1.9]	0.025
Triglycerides, mmol/L, Me [IQR]	1.7 [0.9–2.4]	1.0 [0.7–1.3]	0.085
Increase in TG, *n* (%)	10 (50.0)	17 (89.5)	0.017
Presence of dyslipidemia, *n* (%)	25 (68.8)	16 (73.7)	0.53
C-reactive protein, mL/L, Me [IQR]	1.7 [1.1–2.57]	1.22 [0.66–3.01]	0.75
Increase in CRP, *n* (%)	3 (11.0)	0	0.63
Fibrinogen, g/L, Me [IQR]	4 [3.1–5.8]	3 [2.6–3.9]	0.12
Increase in fibrinogen, *n* (%)	5 (18.0)	1 (4.0)	0.034
NT-proBNP, pg/mL, Me [IQR]	104 [60–121]	90 [52–116]	0.8
Increase in NT-proBNP, *n* (%)	5 (18.0)	6 (21.0)	0.851
Fasting plasma glucose, mmol/L, Me [IQR]	8.23 [6.75–10.5]	5.6 [4.7–6.3]	<0.001
Glycated hemoglobin, %, Me [IQR]	6.5 [6.0–7.2]	5.3 [5.1–5.7]	<0.001
Insulin, µU/mL, Me [IQR]	13.6 [8–22.3]	8.0 [5.7–11.0]	0.007
C-peptide, ng/mL, Me [IQR]	2.74 [2.05–3.75]	1.66 [1.5–2.3]	0.013
NOMA Index, Me [IQR]	6 [3–9]	2 [1–3]	<0.001
Availability of IR, *n* (%)	25 (87.5)	9 (32.0)	0.004

Values are given as *n* (%, in round brackets); median, interquartile range (25th percentile to 75th percentile, in square brackets) or mean ± standard deviation (without brackets)

**Table 2 biomedicines-12-02031-t002:** Structural and functional echocardiographic parameters at baseline and after 7 years of follow-up according to diabetic status.

Variable	Patients with T2D		*p* Value	Patients without T2D		*p* Value
	Visit 1	Visit 2		Visit 1	Visit 2	
LVD, *n* (%)	27 (53)	18 (61)	0.004 *	28 (39)	9 (32)	0.48
LV mass index, g/m^2^	95 [84.1–110]	90.5 [82.0–110.5]	0.49	84.8 [73–98.3]	88.0 [84.0–100.0]	0.84
LA volume index, mL/m^2^	27.0 [24.2–31.1]	26.5 [23.2–33.0]	0.63	26 [23.9–27.8]	25.0 [21.0–33.0]	0.66
IVS in diastole, cm	1.2 [1.1–1.2]	1.25 [1.15–1.4]	0.046	1.0 [1.0–1.2]	1.2 [1.0–1.2]	0.06
Posterior Wall thickness, cm	1.0 [1.0–1.0]	1.1 [1.0–1.15]	0.007	0.93 [0.9–1.0]	0.9 [0.9–1.0]	0.53
Relative Wall Thickness	0.44 [0.41–0.48]	0.49 [0.45–0.53]	0.06	0.43 [0.41–0.46]	0.45 [0.43–0.49]	0.58
Concentric Hypertrophy, *n* (%)	41 (22)	15 (53)	0.025	26 (12)	5 (16)	0.15
Concentric Remodeling, *n* (%)	28 (15)	12 (42)	0.063	33 (16)	6 (20)	0.65
Transmitral E/A ratio	0.8 [0.7–0.9]	0.7 [0.6–0.9]	0.03 *	1.1 [0.9–1.2]	1.2 [0.7–1.3]	0.15
Isovolumic relaxation time, ms	91 [87–96]	83 [72–99]	0.49	77 [70–88]	74 [67–95]	0.83
Deceleration time, ms	218 [187–232]	226 [191–277]	0.05 *	181 [169–203]	183 [169–229]	0.53
Average (med/lat) e’, cm/s	8.7 [7.1–9.6]	6.7 [5.6–8.7]	0.034 *	10.2 [8.3–13.6]	9.0 [8.0–12.6]	0.059
E/e’	7.2 [6.4–8.5]	8.7 [7.9–11.1]	0.017 *	6.4 [5.4–7.6]	7.0 [5.6–8.8]	0.42
Ejection Fraction, %	60 [59–62]	65 [58–79]	0.088	64 [61–66]	67 [60–71]	0.48
GLS, %	−17.6 ± 1.4	−17.2 ± 1.24	0.008 *	−19.6 ± 1.8	−19.6 ± 1.8	0.26

Values are given as described in Table 1’s footer; *, statistically significant *p*-value; LVD, left ventricular dysfunction; LV, left ventricle; LA, left atrium; IVS, interventricular septum; E, early diastolic transmitral flow velocity; A, late diastolic transmitral flow velocity; e’, early diastolic tissue velocity; GLS, global longitudinal strain.

## Data Availability

Data will be made available upon request.

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
