# Peer review of "Subclinical Left Ventricular Dysfunction over Seven-Year Follow-Up in Type 2 Diabetes Patients without Cardiovascular Diseases"

_biomedicines, 2024, doi:10.3390/biomedicines12092031_

Round 1

Reviewer 1 Report

Comments and Suggestions for Authors

An interesting work of important clinical significance is presented for review. The authors studied long-term structural/functional disorders of the left ventricular myocardium in patients with T2D without cardiovascular diseases.In general, the work makes a favorable impression, but there are a number of questions and wishes about the work: 1. It is necessary to give a transcript of the abbreviations when they are first used (for example LV, line 11); 2. in the relevance of the work, data on the prevalence of the studied disorders should be provided; 3. It is necessary to significantly expand the relevance of the work and present in this section data on the medical and social significance of the studied disorders, their consequences, the importance of their prediction and timely correction, etc. 4. It is not clear why the justification of the relevance of the work is based on only one literary source? Has only one work been previously done on this topic? It is necessary to significantly expand the list of modern literature used to substantiate this work. 5. Were any covariates used in the statistical analysis of the data, since the studied samples differ from each other (for example, by BMI, Table 1)? How was the list of covariates used justified?

Author Response

Comments 1. It is necessary to give a transcript of the abbreviations when they are first used (for example LV, line 11).

Response 1: Thank you for pointing this out. We agree with this comment and have made change.

Comments 2.  In the relevance of the work, data on the prevalence of the studied disorders should be provided;

Comments 3. It is necessary to significantly expand the relevance of the work and present in this section data on the medical and social significance of the studied disorders, their consequences, the importance of their prediction and timely correction, etc

Response 2, 3: Thank you for pointing this out. We agree that the “Introduction” section is scant and insufficient.  Since comments 2 and 3 both concern this section, we thought it possible to give one response to them. We have almost completely updated this section, responding to all points of your comments. All additions and changes are highlighted in red: p.1-2; paragraph 1; lines 29-32, 38-77; p.15-18 References.

Comments 4. It is not clear why the justification of the relevance of the work is based on only one literary source? Has only one work been previously done on this topic? It is necessary to significantly expand the list of modern literature used to substantiate this work. 

Response 4:  Thank you for pointing this out. We agree with this comment. We have significantly expanded the number of modern literary sources substantiating our work. The citation and the discussion of these works are included in the “Discussion” section. All additions are highlighted in red: p.13-14; paragraph 4; lines 371-391, 407-418; p.15-18; References.

Comments 5. Were any covariates used in the statistical analysis of the data, since the studied samples differ from each other (for example, by BMI, Table 1)? How was the list of covariates used justified?

Response 5: All potential risk factors (gender, age, BMI, stage 1 obesity, exercise tolerance, lipid levels, carbohydrate metabolism parameters, etc. - see Table 1) were included one by one in the univariate analysis with development DD (diastolic disfunction) and GLS (global longitudinal strain). According to its results, variables that showed p<0.1 in the univariate analysis were included in the model. Then, in the multivariate analysis, they were step-by-step excluded from the equation using the Wald method backwards, and what remained was entered into the graph (Fig.4 for DD, Fig.5 for GLS). Variables with a p value <0.05 were selected as independent predictors from the final model. The results of multivariate analysis are presented as odds ratio (OR) and 95% confidence interval (95% CI). Additions have been made to section 2.5 “Statistical analysis” (p.4, lines highlighted in red 163-167) and to section 3.2 “Echocardiography” (p.10, lines highlighted in red 263-266).

In conclusion, we are very grateful to you for your great job in reviewing our manuscript. We hope that your comments helped us improve the article.

Reviewer 2 Report

Comments and Suggestions for Authors

After reading the manuscript “Subclinical Left Ventricular Dysfunction over Seven-year Follow-up in Type 2 Diabetes Patients without Cardiovascular Diseases” by Dariga Akasheva and all. that aimed to establish the long-term structural and functional disorders of the left ventricle, I have some comments:

-      -    The introduction is too brief. Maybe some additional information about the reasons that made you choose this topic for research.

-     -     The number of patients is too small, and the statistical results can be influenced by the number of patients

Author Response

Сomments 1. The introduction is too brief. Maybe some additional information about the reasons that made you choose this topic for research.

Response 1: Thank you for pointing this out. We agree that the “Introduction” section is scant and insufficient. We have almost completely updated this section with additional information. All additions are highlighted in red: p.1-2; paragraph 1; lines 29-32, 38-77; p.15-18 References.

Comments 2. The number of patients is too small, and the statistical results can be influenced by the number of patients.

Response 2. We certainly agree with this remark. The article provides a detailed justification for this limitation. I will list it point by point: 1) the initial small number of participants; 2) the usual reluctance of many people to participate in a repeat study after several years; 2) the height of the covid epidemic. Nevertheless, after analyzing the obtained results, we decided to publish them despite the small number of study participants. The statistical results are of course affected by the number of patients. The more patients, the more reliable the results. However, the results of our statistical analysis also contributed to our decision to publish the data obtained.

In conclusion, we are very grateful to you for your work in reviewing our manuscript. We hope that your comments helped us improve the article.

Round 2

Reviewer 1 Report

Comments and Suggestions for Authors

The authors have made all reasonable adjustments and provided answers to all questions. The article is recommended for publication.